# Ascarosides and Symbiotic Bacteria of Entomopathogenic Nematodes Regulate Host Immune Response in *Galleria mellonella* Larvae

**DOI:** 10.3390/insects15070514

**Published:** 2024-07-09

**Authors:** Kanjana Chantab, Zhongchen Rao, Xuehong Zheng, Richou Han, Li Cao

**Affiliations:** 1Guangdong Key Laboratory of Animal Conservation and Resource Utilization, Guangdong Public Laboratory of Wild Animal Conservation and Utilization, Institute of Zoology, Guangdong Academy of Science, Guangzhou 510260, China; kanjana.ke@rmuti.ac.th (K.C.); raozc@giz.gd.cn (Z.R.); zhengxh@giz.gd.cn (X.Z.); hanrc@giabr.gd.cn (R.H.); 2Department of Plant Sciences, Faculty of Agriculture and Technology, Rajamangala University of Technology Isan, Surin 32000, Thailand

**Keywords:** entomopathogenic nematodes, *P. luminescens* subsp. *kayaii* H06, ascarosides, immunity, color change, mortality

## Abstract

**Simple Summary:**

Entomopathogenic nematodes and their symbiotic bacteria are widely utilized for biological control. However, whether nematode-derived signals influence the immune response of insects remains unexplored. In this study, we provide comprehensive insight into the interactions among symbiotic bacteria, nematode pheromone ascarosides, and the *G. mellonella* insect host. Our findings reveal that under the induction of *Photorhabdus* bacteria, ascarosides enhance the host’s immunity response. Specifically, they suppress the intensity of body color change and increase the expression of related-immunity gene expression. Additionally, ascarosides reduce the *Photorhabdus* cell load and delay *G. mellonella* larval mortality. These results imply that *G. mellonella* larvae may employ nematode pheromones to enhance insect immunity in the presence of symbiotic bacteria, thereby enhancing the resistance to invasive bacteria.

**Abstract:**

Insects protect themselves through their immune systems. Entomopathogenic nematodes and their bacterial symbionts are widely used for the biocontrol of economically important pests. Ascarosides are pheromones that regulate nematode behaviors, such as aggregation, avoidance, mating, dispersal, and dauer recovery and formation. However, whether ascarosides influence the immune response of insects remains unexplored. In this study, we co-injected ascarosides and symbiotic *Photorhabdus luminescens* subsp. *kayaii* H06 bacteria derived from *Heterorhabditis bacteriophora* H06 into the last instar larvae of *Galleria mellonella*. We recorded larval mortality and analyzed the expressions of AMPs, ROS/RNS, and LPSs. Our results revealed a process in which ascarosides, acting as enhancers of the symbiotic bacteria, co-induced *G. mellonella* immunity by significantly increasing oxidative stress responses and secreting AMPs (*gallerimycin*, *gloverin*, and *cecropin*). This led to a reduction in color intensity and the symbiotic bacteria load, ultimately resulting in delayed host mortality compared to either ascarosides or symbiotic bacteria. These findings demonstrate the cross-kingdom regulation of insects and symbiotic bacteria by nematode pheromones. Furthermore, our results suggest that *G. mellonella* larvae may employ nematode pheromones secreted by IJs to modulate insect immunity during early infection, particularly in the presence of symbiotic bacteria, for enhancing resistance to invasive bacteria in the hemolymph.

## 1. Introduction

The greater wax moth, *Galleria mellonella* (L.) (Lepidoptera: Pyralidae), is increasingly utilized in scientific research due to its ease of reproduction, numerous offspring, and short development cycle [1]. Its well-known immune-related transcriptome provides a convenient research model for investigating insect immunity at biochemical and molecular levels [1]. During the infection of pathogens, insects secrete defense molecules, recognize pathogens, induce signal transduction, eliminate microbial debris, and regulate transcription factors for immune effector expression. The recognition of specific molecules through specialized pattern recognition receptors (PRRs) plays a central role in insect immune responses [2,3,4]. Insects utilize PRRs to detect various molecular signatures associated with microbes, activating innate immune responses by secreting antibacterial molecules, such as reactive oxygen species (ROS), reactive nitrogen species (RNS), and/or antimicrobial peptides (AMPs) [5]. The innate immune defense is bolstered by the production of antimicrobial ROS and RNS. The humoral defenses of the greater wax moth include at least 18 antimicrobial peptides, especially gallerimycin, gloverin, and galiomycin [6,7,8]. These antimicrobial peptides are produced broadly in epithelial cells and specialized cells. Insects simultaneously produce multiple AMPs, enhancing their antimicrobial effect and serving as key components of innate defense responses, due to the absence of adaptive immune responses [9,10].

*Steinernema* and *Heterorhabditis* spp., entomopathogenic nematodes (EPNs), are respectively associated with *Xenorhabdus* or *Photorhabdus* symbiotic bacteria in the intestine during the infective juveniles (IJs) stage. These EPNs serve as excellent bioinsecticides for controlling various economically important insect pests [11,12,13]. They rapidly infect and kill insect hosts within the first few days of infection [14]. The nematodes penetrate the host’s natural openings or cuticles and release their endosymbiotic bacteria into the host’s hemolymph, where these bacteria produce virulence factors contributing to the host’s demise. In response, the insect host activates defense pathways similar to those induced by other pathogens [15]. Upon entering the insect hemolymph, these nematodes employ molecular mimicry and modulation of the host’s immune response as their primary strategies to evade the host’s immune system [16]. During early infection stages, nematodes mimic host microbiota, preventing recognition by displaying surface molecules that interfere with hemocyte surveillance. This allows the host to distinguish between symbionts and pathobionts [17], a crucial distinction in disease research, particularly when pathobiont levels increase [3]. Notably, Gram-negative bacteria regulate lipopolysaccharide (LPS)-binding molecules, including apolipophorin and hemolin, in *G. mellonella* larvae. When pathobionts are administered to *G. mellonella* larvae, there are stronger gene expressions of LPS-binding molecules, ApoIII, and hemolin, compared with the administration with the symbiont [18], indicating a significant influence on the host’s immune mechanisms.

From an evolutionary standpoint, infective juveniles (IJs) of EPNs must become activated, undergo morphological changes, and consume the food resources within the invaded insect host. This process relies on the release of virulent factors and necessary signals to complete the nematode life cycle. The excretion and secretion products of EPNs may be linked to host immunosuppression during the invasive initial stage, promoting a suitable growing environment for the released symbiotic bacteria [19,20]. Nematode infection also induces stress-signaling cascades that result in the synthesis of nitric oxide (NO) and the differential regulation of genes involved in reactive oxygen species (ROS) production [21,22]. The host’s defense response likely occurs due to the recognition of conserved nematode signature molecules, such as ascarosides. However, whether ascarosides induce an insect’s immune response has not been reported in earlier studies.

Ascarosides are glycosides of ascarylose, containing a fatty acid-derived lipophilic side chain and optionally decorated with additional metabolic building blocks [23]. More than 200 ascaroside structures secreted by over 20 nematode species have been identified, playing a crucial role in nematode chemical communication and regulating various aspects of their communication and social behaviors [24,25,26,27,28,29,30,31]. While nematode pheromones primarily function between the same species, recent research has revealed their effects across different species, including fungi, plants, and insects [32,33,34]. For instance, the nematode pheromone ascaroside ascr#18 activates plant defenses gene expression and provides protection against a broad spectrum of pathogens [32]. Plants use chemical signaling to modify nematode pheromones into shorter, side-chained ascarosides, which act as a defense mechanism alongside conventional pattern-triggered immunity [33]. Interestingly, in various eukaryotes and some bacteria, ascarosides can be converted into derivatives with a shortened fatty acid side, similar to ascaroside biosynthesis in nematodes [35]. However, whether nematode-derived signals influence the immune response of insects remains unexplored. The aim of this research was to examine how the *G. mellonella* larval immune system is modulated in response to EPNs’ signal ascarosides and symbiotic bacteria. Two ascarosides (ascr#9 and ascr#12), together with *P. luminescens* subsp. *kayaii* H06 bacteria, were co-injected into the last instar larvae of *G. mellonella*. Larval mortality was recorded, and the expressions of the genes related to AMPs, ROS/RNS, and LPSs were analyzed.

## 2. Materials and Methods

### 2.1. Insects and Symbiotic Bacteria Culture

The last instar larvae of *G. mellonella* reared on an artificial diet were used for this study. The healthy larvae were kept in plastic containers at 25 °C for 2 days in the laboratory to prevent interference from background stress.

The symbiotic bacterium *P. luminescens* subsp. *kayaii* H06 [36] was cultured on a 120 rpm shaker at 25 °C in 100 mL of Luria–Bertani (LB) medium for 24 h, and 10 mL of bacterial suspension (OD = 0.4) was subcultured overnight in a 250 mL flask containing 100 mL of LB medium. The bacterial cells were harvested, cleaned by centrifuging at 12,000 rpm for 10 min at 10 °C using sterile 1xPBS (pH = 7.0), and stored in sterile 1xPBS for immediate use. Cell counts were performed under a microscope, and the suspension was adjusted to 10^8^ cells/mL with sterile 1xPBS.

### 2.2. Ascarosides

Two most abundant ascarosides, ascr#9 (asc-C5) and ascr#12 (asc-C6), based on the identified chemical structures found in the *Heterorhabditis* nematode, were used for injection. These ascarosides were synthesized with a purity of over 95% by WuXi AppTec, Tianjin, China and kept in a temperature of −20 °C [31]. The ascarosides were diluted to the working concentrations with sterile PBS prior to experimental use.

### 2.3. Injection of G. mellonella Larvae

Under the clean cabinet, *G. mellonella* larvae were washed twice with sterile extra-pure water, and their body surface was dried using sterile filter paper. The larvae were then immobilized on ice for 15 min before injection into the pro-leg, with each treatment using the microinjection system (IM-31; Narishige, Tokyo, Japan). Each larva received 4 μL of suspension containing one of the following: sterile PBS, a mixture of ascr#9 and ascr#12 (equal volume) at 4 mM, bacterial cells (2 × 10^8^/mL), a mixture of ascarosides (ascr#9 and ascr#12 with equal volume at 6 mM), and bacterial cells (6 × 10^8^/mL). Non-injected larvae served as the negative control. In total, 60 larvae were used for each treatment in each sampling time: 20 larvae for qRT-PCR and 40 larvae for mortality assay (with 4 replicates). Two sampling times were conducted for each treatment. After injection, the larvae were placed in the 24-well cell plates at 25 °C for data sampling and analysis. The experiments were conducted twice.

### 2.4. Larval Mortality and Color Intensity Change

After injection, the larval mortality was monitored at 12 h intervals until all the larvae died. Additionally, after 12 h of treatment, the dead larva body was examined visually to monitor the formation of melanin intensity. The degree of body color change was classified according to [37]: none (no visible signs of melanin), light (visible translucent melanin), moderate (melanin with pigment, semi-opaque to fully opaque), and high (melanin with strong pigmentation). The experiment was conducted in duplicate, with 4 replications for each treatment.

### 2.5. RNA Extraction and cDNA Preparation

The larvae were incubated after injection for 4 and 10 h. The larvae samples were frozen in liquid nitrogen and homogenized with TRIzol reagent (Invitrogen, Waltham, MA, USA). The homogenates were mixed with 200 μL of trichloromethane, and the supernatant was collected by centrifugation at 12,000× rpm for 15 min at 4 °C. Following centrifugation, the upper layer was transferred to a new tube, and RNA was washed with ethanol. RNA concentration and quality were checked by a UV spectrophotometer (NanoDrop Technologies, Thermo Scientific, Wilmington, DE, USA). The RNA concentration was normalized to 1 μg/μL. cDNA was generated with the PerfectStartTM Green qPCR SuperMix Kit (TransGen, Beijing, China) and diluted tenfold with pure water before qRT-PCR analysis.

### 2.6. qRT-PCR Analysis

Gene expression analysis by qRT-PCR was conducted using a CFX96 Real-Time System and a C1000 Thermal Cycler (Bio-Rad, Hercules, CA, USA). Three biological replicates with 5 larvae in each replicate were set up for each treatment. The reactions consisted of 10 μL of qPCR Mix (TAKARA, Shiga, Japan), 40 ng of cDNA template, forward and reverse primers at a final concentration of 100 nM, and water at a final volume of 20 μL. The cycling conditions were as follows: 95 °C for 30 s, 40 repetitions of 95 °C for 5 s, followed by 60 °C for 30 s, and then one round of 95 °C for 15 s, 65 °C for 5 s, and finally, 95 °C for 5 s. Fold change was calculated using the 2^−ΔΔCt^ method, with all values normalized to the expression of the reference gene GAPDH. Primer sets for gallerimycin, gloverin, and cecropin, together with the LPS-binding gene hemolin and ROS/RNS-related genes (GST, Nos, and Nox), were checked for gene specificity using PCR and agarose gel electrophoresis (Appendix A)

### 2.7. Amplicon Sequencing

Amplicon sequencing was employed to investigate the bacterial microbiota of the larval hemolymph from the injected and non-injected larvae. Total DNA from each sample was extracted and purified using the QIAamp DNA Stool Mini Kit (Qiagen, Hilden, Germany) according to the manufacturer’s instructions. After quantification by the NanoDrop ND-3300 spectrophotometer (NanoDrop Technologies, Thermo Scientific, Wilmington, DE, USA), PCR was carried out to generate amplicons in quintuplicate for the samples. The V5–V6 region of the bacterial 16S rRNA gene was amplified using primers 799F/1193R [38]. The amplicons were sequenced on an Illumina Nova 6000 platform (Guangdong Magigene Biotechnology, Guangzhou, China). The sequencing data were quality checked with Fastp (v. 0.14.1). Paired-end clean reads were merged using FLASH (v. 1.2.11) according to the relationship of the overlap between the paired-end reads. Raw tags with at least 10 reads overlapping the opposite end of the same DNA fragment were merged (error < ratio 0.1). Sequences were assigned to each sample based on their unique barcodes. The barcodes and primers were removed before the clean tags were generated. Based on the OTUs table, the unique, shared, and core OTUs among four groups were illustrated by the UpSetR package using the R software [39]. The bacterial compositions of the class, family, and genus levels for each group were calculated, and the histogram was drawn with the ggplot2 package using the R software (v. 3.3.2).

### 2.8. Statistical Analysis

The data were expressed as the mean + standard error (SE). Statistical analysis was performed using the SPSS 16.0 software (SPSS, Chicago, IL, USA). The datasets were analyzed using the one-way analysis of variance (ANOVA) model, followed by the least-squared difference (LSD) test. Differences were considered statistically significant if the *p*-value was less than 0.05 (*p* < 0.05).

## 3. Results

### 3.1. Ascarosides Delayed the G. mellonella Larvae Mortality

The results showed that neither PBS nor ascarosides (ascr#9 and ascr#12) significantly influenced the larval survival, compared to the non-treated control. However, symbiotic bacteria caused greater larval mortality within 12 h after injection (Figure 1). All the larvae treated with symbiotic bacteria died by 84 h post-injection. The larvae treated with symbiotic bacteria exhibited a significantly higher mortality (53%) at 72 h after injection, compared to non-treated larvae (2.5%), PBS-treated larvae (2.5%), or ascaroside-treated larvae (2.5%). Co-injection of ascarosides and bacterial cells resulted in a 35% mortality (*F* = 41.16, *df* = 4, 15, *p* < 0.01). Notably, larvae treated with both ascarosides and bacterial cells showed a 30% decrease in mortality, compared to those treated with bacterial cells alone (Figure 1 and Appendix A). This finding suggests that ascarosides can delay larval mortality in the presence of symbiotic bacterial cells.

### 3.2. Ascarosides Reduce the Degree of Color Change in G. mellonella Larvae

The color change response of *G. mellonella* larvae to symbiotic bacteria cells was observed 12 h after injection. The degrees of color change were classified as none, light, moderate, or high, based on the intensity of black pigment melanin in the body (Figure 2A). The body color change was recorded within the first few hours after injection. There was no significant difference in body color change intensity between the non-treated group and larvae treated with ascarosides or PBS (*F* = 2.18, *df* = 4, 15, *p* > 0.05) (Figure 2B). However, significant differences in larval body color change response were observed between the non-treated group, PBS-, ascarosides-, and symbiotic bacteria-treated groups. These differences were categorized as follows: no color change (*F* = 49.50, *df* = 4, 15, *p* < 0.001), moderate (*F* = 5.46, *df* = 4, 15, *p* < 0.01), and high density (*F* = 3, *df* = 4, 15, *p* < 0.05) (Figure 2B). Symbiotic bacteria or the co-injection of ascarosides and bacterial cells significantly induced the larval body color change (Figure 2B). Notably, the larval numbers with high color change density were significant between symbiotic bacteria and the co-injection of ascarosides and bacterial cells (LSD test; *F* = 3, *df* = 4, 15, *p* < 0.05). This finding suggests that ascarosides reduced the larval color change intensity caused by bacterial cells.

### 3.3. Ascarosides and Symbiotic Bacteria Activated Antimicrobial Peptide Expression

The transcriptional expressions of *gallerimycin*, *gloverin*, and *cecropin* genes in *G. mellonella* larvae were examined specifically at 4 h and 10 h post-injection in response to symbiotic bacteria cells and/or ascarosides. The expressions of the *gallerimycin*, *gloverin*, and *cecropin* genes were significantly up-regulated in the larvae injected with ascarosides and bacterial cells at 4 h and 10 h post-injection (except cecropin at 4 h) and in those with symbiotic bacteria cells at 10 h post-injection (Figure 3). The gene expression levels were markedly higher at 10 h, compared to 4 h. Furthermore, significant up-regulations of *gloverin* and *gallerimycin* genes occurred in the larvae treated with PBS, bacterial cells, and both bacterial cells and ascarosides, compared to non-treated larvae at 4 h. However, individual larvae injected with symbiotic bacteria exhibited a significantly lower gene expression (Figure 3). There was no significant difference in the expressions of *gallerimycin* and *cecropin* between larvae treated with symbiotic bacteria (*F* = 45.73, *df* = 4, 15, *p* > 0.05) and with the co-injection with ascarosides and bacterial cells (*F* = 2.50, *df* = 4, 15, *p* > 0.05) at 10 h. Nevertheless, significant differences were observed in *gloverin* expression at 4 h and 10 h (*F* = 9.84, *df* = 4, 15, *p* < 0.05), as well as in *gallerimycin* expression at 4 h (*F* = 6.27, *df* = 4, 15, *p* < 0.01). There results indicate that physical injury also induced immune gene expression, and the co-injection of ascarosides and bacterial cells strongly activated antimicrobial gene expression in *G. mellonella* larvae.

### 3.4. Hemolin Was Up-Regulated in Response to Ascarosides and Symbiotic Bacteria

*Photorhabdus* symbiotic bacteria, being Gram-negative, contain lipopolysaccharides (LPSs) that are recognized by the *G. mellonella* larvae. The regulation of the LPS-binding molecule hemolin in *G. mellonella* larvae was observed. The larvae injected with PBS or ascarosides did not significantly induce hemolin expression, compared to the untreated control (Figure 4). However, the administration of both ascarosides and bacterial cells significantly stimulated hemolin gene expression in the larvae at 4 h, compared to symbiotic bacteria cells alone (*F* = 9.32, *df* = 4, 15, *p* < 0.05) and other treatments (*F* = 9.32, *df* = 4, 15, *p* < 0.001). Interestingly, no significant difference in hemolin expression was observed after 10 h (Figure 4).

### 3.5. ROS/RNS-Related Gene Expression Increased in Response to Symbiotic Bacteria and Ascarosides

The impact of ascarosides and bacterial cells on the expression of the ROS/RNS-related genes (*GST*, *Nos*, and *Nox*) was determined in *G. mellonella* larvae. Larval administration of both bacterial cells and ascarosides significantly up-regulated ROS/RNS-related genes, compared to symbiotic bacteria alone (*F* = 34.20, *df* = 4, 15, *p* < 0.01)/(*F* = 3.78, *df* = 4, 15, *p* < 0.01) or ascarosides alone, at 10 h ((*F* = 34.20, *df* = 4, 15, *p* < 0.01)/(F(4, 15) = 3.784, *p* < 0.01)) (Figure 5). Symbiotic bacteria induced *GST* and *Nos* gene expression in the larvae after 4 and 10 h but did not induce *Nox* expression at 4 h (Figure 5). The expression of Ros/RNS-related genes in the larvae treated with PBS or ascarosides was less pronounced than in larvae exposed to bacterial cells.

### 3.6. Ascarosides and Symbiotic Bacteria Decrease Symbiotic Bacterial Load

The larval mortality was tardy when co-injection with ascarosides and bacterial cells were treated with the larvae, as described in the above results. Therefore, the influence of ascarosides on the growth of symbiotic bacteria at 4 h and 10 h was investigated. Using qRT-PCR targeting the *P. luminescens* subsp. *kayaii* 16S rRNA and amplicon analysis, we detected a significant reduction in bacteria abundance in the larval hemolymph following the administration of both ascarosides and bacterial cells. This resulted in an approximately fourfold decrease in transcript expression, compared to bacterial cells alone (*F* = 10.97, *df* = 4, 15, *p* < 0.01). Notably, no 16S rRNA expression was detected in the control larval hemolymphs treated with PBS or ascarosides, or that received no treatment (Figure 6A,B). Amplicon sequencing analysis further confirmed that operational taxonomic units (OTUs) of *Photorhabdus* bacteria in the larval hemolymph were significantly decreased by 37% and 56% at 4 and 10 h, respectively, when both ascarosides and bacterial cells were administered. This reduction was compared to larvae treated with *Photorhabdus* cells alone (*F* = 9.19, *df* = 4, 10, *p* < 0.05) (Figure 6C,D). Collectively, these results suggest that the presence of ascarosides retards the growth of symbiotic bacteria, as determined by qRT-PCR and amplicon analysis.

## 4. Discussion

In this study, a comprehensive characterization of the intensity of body color change and humoral immunity toward pheromone ascarosides and symbiotic *Photorhabdus* bacteria in the *G. mellonella* model was provided. In this study, we demonstrated that the co-injection of ascarosides (ascr#9 and ascr#12) and bacterial cells into *G. mellonella* larvae delayed the larval mortality, suppressed larval body color intensity, up-regulated immune-related genes and ROS/RNS-related genes, and decreased *Photorhadus* abundance, especially during the early stages of infection. The cross-kingdom regulation of insects and symbiotic bacteria associated with entomopathogenic nematodes by nematode pheromones was also demonstrated.

The ingestion of hemocytes by *C. elegans* within the hemocoel of *G. mellonella* represents a significant finding, revealing a novel mechanism by which bacterial-feeding nematodes evade host immune responses [40]. The temporally dynamic expression of a subset of antimicrobial peptide genes in *Drosophila* larvae is activated by *P. luminescens* but not by axenic *Heterorhabditis* nematodes [41]. This suggests that axenic *Heterorhabditis* nematodes may escape the *Drosophila* immune response through unknown factors. While it is reasonably hypothesized that axenic *Heterorhabditis* nematodes may produce signals such as ascarosides to protect themselves by inhibiting the larval antimicrobial peptide genes, the presence of ascarosides within *Photorhabdus* markedly stimulates the expressions of immune- and ROS/RNS-related genes and LPS-binding *hemolin* in *Gallleria* larvae, compared to the presence of ascarosides or *Photorhabdus* alone. The study by Roder et al. (2019) [42] sheds light on the fascinating dynamics between *Steinernema* nematodes, their symbiotic bacteria, and ascaroside signaling. The differential impact of the *Xenorhabdus nematophila* All strain on ascaroside production in *S. feltiae*, compared to *S. carpocapsae*, highlights the complexity of this interaction.

Insects rely on pattern recognition receptors (PRRs) to sense microbial components, such as LPSs, initiating appropriate mechanisms to eliminate potential invaders [43]. Gram-negative bacteria release LPSs, which bind to extracellular proteins, resulting in immune signaling [44]. In *G. mellonella*, apolipophorin and hemolin serve as identified LPS-binding proteins [18]. *G. mellonella* induces immune responses with varying intensities against different microbes [45]. The insect immune system recognizes the presence of *P. luminescens* subsp. *kayaii* H06 at both early and late infection stages [4]. In *M. sexta* infected with *Photorhabdus*, hemolin expression is typically triggered, along with the peptidoglycan recognition protein [46]. *Hemolin*, abundant in insect hemolymph is proposed as a recognition protein that interacts with LPS [47]. Our results indicate a significantly up-regulated *hemolin* expression in the larvae treated with both ascarosides and bacterial cells at 4 h. Entomopathogens employ evasion and interference strategies during early infection stages, interacting with host receptors for immunological discrimination between self and non-self. These interactions can modify the pathogen into an immunocompatible structure specific to host factors [48]. It appears that nematode ascarosides signaling can penetrate the epithelial cell barrier, enabling the host to recognize the invasive pathogen through the host LPS-binding protein.

Insects produce AMPs, as well as ROS and RNS, upon microbial challenge [49]. However, no reports on insect immune responses to entomopathogenic nematode (EPN) signals are currently available. ROS and RNS play critical roles in inhibiting microbial growth and maintaining intestinal homeostasis [50]. DUOX-related signaling is involved in balancing the intestinal microbiota load [51]. A total of 18 known AMPs in different classes that have been identified in *G. mellonella*, gallerimycin, gloverin, and cecropin are frequently investigated [52,53,54]. In this study, symbiotic bacteria induced significant expressions of three AMPs, *gallerimycin*, *gloverin*, and *cecropin*, in the presence of ascarosides. Additionally, our results demonstrated that the up-regulations of *GST*, *Nox*, and *Nos* in *G. mellonella* may be induced by both ascarosides and bacterial cells, contributing to a reduction in the *Photorhabdus* bacterial load, as determined by qRT-PCR and amplicon analysis. It is worth noting that the expression of these genes may also inhibit other commensal bacteria in the larval hemolymph. Different nematode receptors responding to various ascarosides have been identified [55,56]. How *G. mellonella* larvae perceive the nematode signals is worthy of further research.

Melanization (body color change) is a crucial process in insect humoral responses, responsible for forming melanin coats around foreign bodies as a defense against non-self elements in arthropods [18]. When infected by pathogens, insects initiate a polyphenol oxidase cascade, resulting in melanization within a few minutes [57]. Interestingly, secretions from *S. carpocapsae* nematodes decrease the melanization response and hemocyte levels but lead to increased mortality, due to the high expression of antimicrobial peptide genes in male *Drosophila melanogaster* flies [37]. On the other hand, *X. nematophila* inhibits the phenoloxidase cascade, resulting in a low melanization level while activating antimicrobial peptides synthesis in *D. suzukii* larvae [58]. Both *Xenorhabdus* and *Photorhabdus* produce specific inhibitors of phospholipase, a key component of melanization. This suggests that the bacterial endosymbionts associated with EPNs promote nematode survival by suppressing melanization reactions [59,60]. The results of this study indicate that the administration of ascarosides reduced *Photorhabdus*-induced color change intensity in the larvae. This finding aligns with the previous research showing that *S. carpocapsae* associated with the *Xenorhabdus nematophila* induces the expression of a subset of AMPs and suppresses melanization [61].

Although the first ascaroside-based signaling molecules were identified in the free-living model *Caenorhabditis elegans* [26,62], ascarosides have been found in organisms across multiple kingdoms beyond nematodes, such as nematophagous fungi, which set traps to capture and digest nematodes and also produce ascarosides [63]; monocot and dicot plants have been identified as sources of ascarosides [64], which is extensively metabolized by a wide range of phyla, including plants, fungi, bacteria, mammals, and nematodes [34]. Interestingly, nematode pheromones can function in different biological kingdoms. Ascarosides contribute to resistance against pathogens, including viruses, bacteria, fungi, oomycetes, and nematodes in different plant crops [32]. For example, the ascaroside ascr#18, secreted by plant-parasitic nematodes, is metabolized by plants to generate chemical signals that repel nematodes and reduce infection [33]. Ascr#7 significantly modulates the pulmonary immune response and reduces asthma severity in mice. It achieves this by suppressing the production of IL-33 from lung epithelial cells and reducing the number of memory-type pathogens in the lungs [65].

Interpretation of our data indicates that immune reactions occur in distinct timeframes, which can be categorized into faster and later responses. After 4 h, *GST*, *Nos*, and *Nox* gene expressions were upregulated, while the AMPs- and LPS-related gene expressions increased at 10 h after administration. Therefore, the production of reactive oxygen and nitrogen species (ROS/RNS) should occur before *hemolin* expression is induced (at 4 h), triggering a cytoprotective survival program that causes oxidative stress. Subsequently, AMPs- and LPS-related gene expressions were induced. Taken together, our results suggest that ascarosides and *Photorhabdus* bacteria induced *G. mellonella* immunity by increasing oxidative stress responses and secreting AMPs, leading to a decreased bacterial load and delayed host mortality (Figure 7). Beetles fed with ascarosides exhibit increased cold tolerance, evidenced by a lower lethal temperature, compared to the control group. This implies that ascarosides enhance the beetles’ cold tolerance and potentially delays mortality under colder environmental conditions [66]. Ascarosides have a multifaceted impact on biocontrol efficacy. They enhance EPN dispersal by influencing nematode movement and the efficient location of insect hosts. Additionally, ascarosides boost EPN infectivity by aiding cuticle penetration and the initiation of infection [67,68]. Leveraging ascarosides directly offers promise for optimizing EPN-based pest management strategies.

## 5. Conclusions

This study explores the interactions between symbiotic bacteria, nematode pheromone ascarosides, and *G. mellonella* insect hosts. It reveals that under the induction of *Photorhabdus* bacteria, ascarosides enhance the host immunity, suppress larval color change intensity, increase related-immunity gene expression, reduce the *Photorhabdus* cell load, and delay *G. mellonella* larval mortality. These results also demonstrate the cross-kingdom regulation of insect immune response and bacterial load by nematode pheromones, implying that *G. mellonella* larvae may employ nematode pheromones secreted by IJs to modulate insect immunity in the presence of symbiotic bacteria for resistance to invasive bacteria in the hemolymph during early infection. The results will help us gain a better understanding of how nematode signals and symbiont bacteria impact the host immune system and potentially lead to the development of innovative pest management strategies for the effective control of insect pests.

## Figures and Tables

**Figure 1 insects-15-00514-f001:**
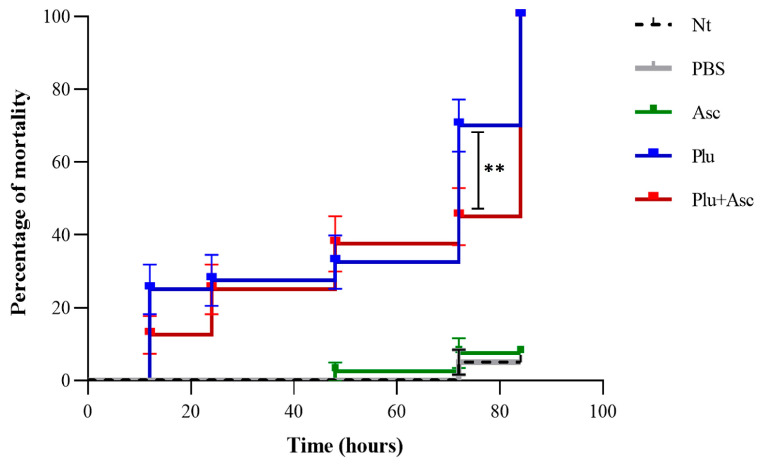
The percentage of *G. mellonella* larval mortality after administration. NT = non-treatment, PBS = injection with sterile PBS, Asc = injection with ascarosides ascr#9 and ascr#12, Plu = injection with symbiotic Photorhabdus bacteria, and Plu + Asc = co-injection with ascarosides and symbiotic bacteria. Significant differences between survival curves were analyzed using ANOVA (** *p* < 0.01).

**Figure 2 insects-15-00514-f002:**
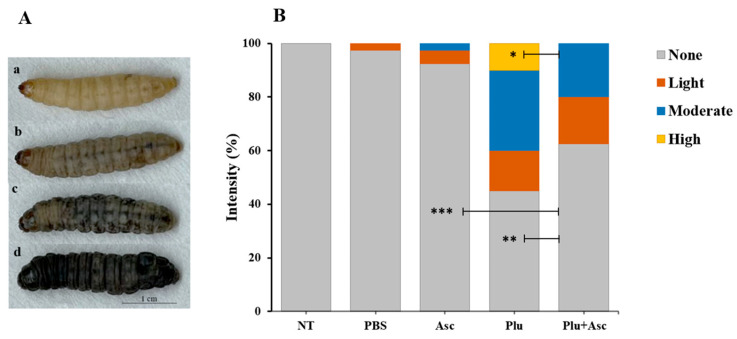
The body color change intensity of *G. mellonella* larvae after administration. The observed color change response in the larvae (**A**) was categorized into four level (**B**) after injection. The abbreviations are given in Figure 1. The data were analyzed using ANOVA (* *p* < 0.05, ** *p* < 0.01, and *** *p* < 0.001).

**Figure 3 insects-15-00514-f003:**
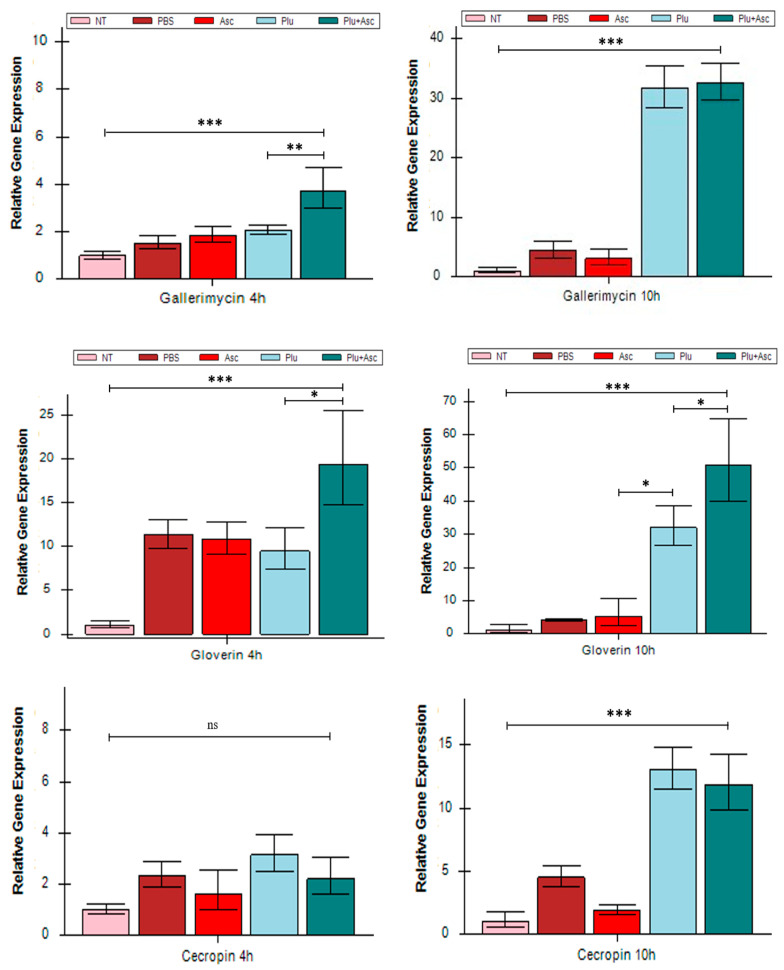
Antimicrobial peptide gene expression in *G. mellonella* larvae after administration. The abbreviations are given in Figure 1. Samples were collected at 4 h and 10 h. The data were analyzed using ANOVA (* *p* < 0.05, ** *p* < 0.01, and *** *p* < 0.001).

**Figure 4 insects-15-00514-f004:**
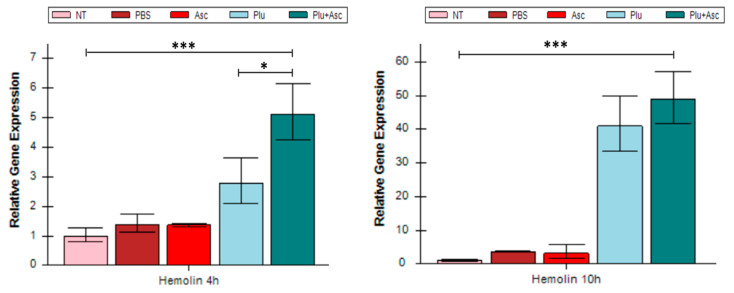
Gene expression of the LPS recognition receptor hemolin from *G. mellonella*. The abbreviations are given in Figure 1. The data were analyzed using ANOVA (* *p* < 0.05 and *** *p* < 0.001).

**Figure 5 insects-15-00514-f005:**
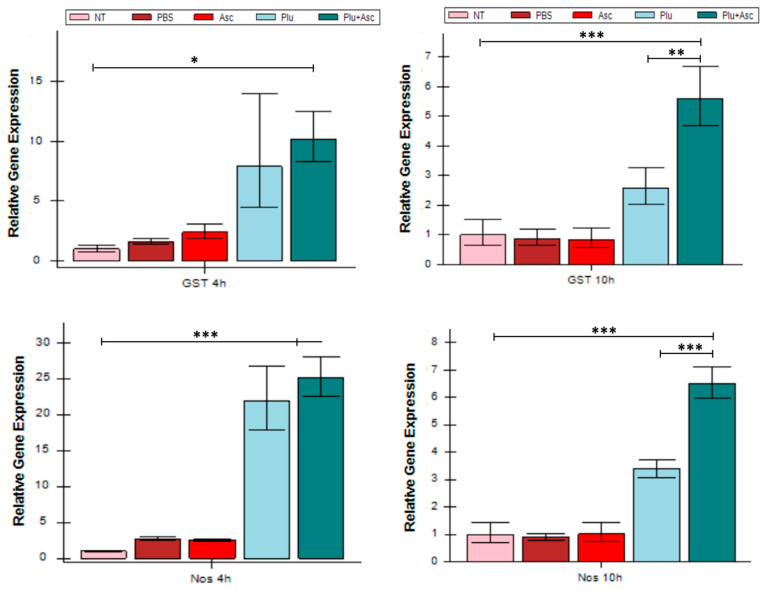
ROS- and RNS-related gene expressions (*GST*, *Nos*, and *Nox*) after administration. The abbreviations are given in Figure 1. Samples were collected at 4 h and 10 h. The data were analyzed using ANOVA (* *p* < 0.05, ** *p* < 0.01, and *** *p* < 0.001).

**Figure 6 insects-15-00514-f006:**
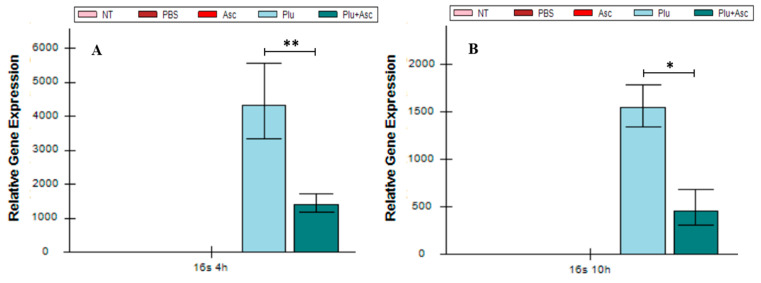
Quantification of the *Photorhabdus* abundance by 16S rRNA at 4 h (**A**) and 10 h (**B**), while amplicon sequencing was also conducted at 4 h (**C**) and 10 h (**D**) from the larval hemolymph after administration. The abbreviations are given in Figure 1. Samples were collected at 4 h and 10 h. The data were analyzed using ANOVA (* *p* < 0.05, ** *p* < 0.01, and *** *p* < 0.001).

**Figure 7 insects-15-00514-f007:**
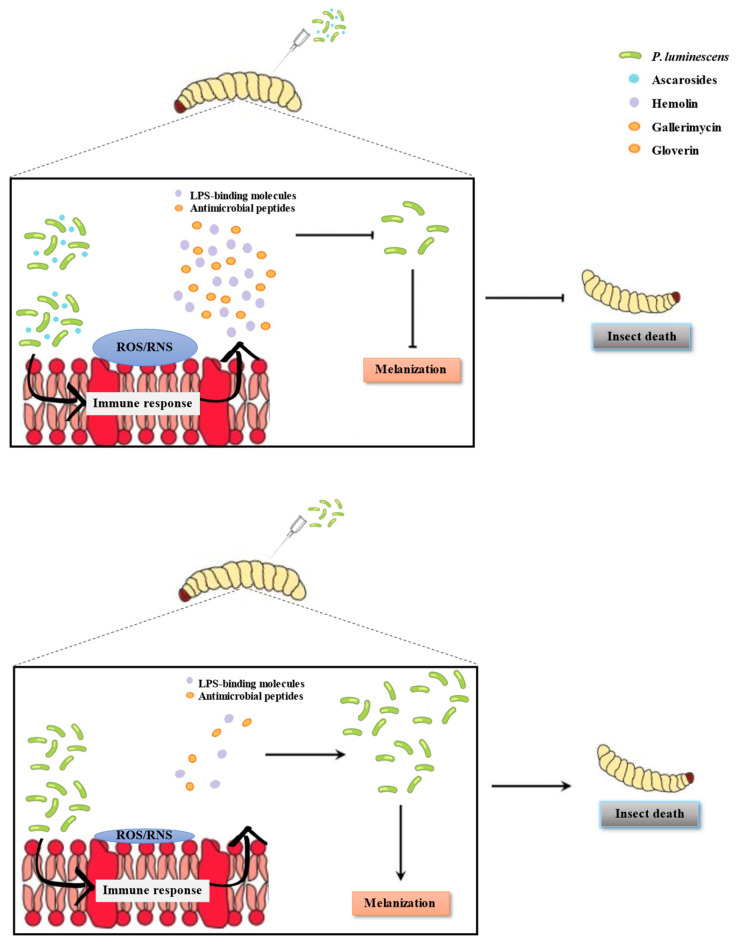
Schematic overview of major results in this study. The results present a process where ascarosides and symbiotic bacteria co-induced *G. mellonella* immunity by significantly increasing oxidative stress responses and secreting AMPs (*gallerimycin*, *gloverin*, and *cecropin*) to reduce color change intensity and the bacterial load, leading to delayed host mortality.

## Data Availability

The data generated in this study are available from the corresponding author upon reasonable request.

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
