# Peer review of "Ascarosides and Symbiotic Bacteria of Entomopathogenic Nematodes Regulate Host Immune Response in *Galleria mellonella* Larvae"

_insects, 2024, doi:10.3390/insects15070514_

Round 1
Reviewer 1 Report
Comments and Suggestions for Authors
This manuscript explores the interaction of ascaroside pheromones in relation to entomopathogenic nematodes (EPNs), their symbiotic bacteria and the insect host immune system (using G. mellonella as a model). The paper is very interesting but needs fairly major revisions prior to publication.
The following issues must be corrected. Some are minor points whereas others are more serious.
· English language writing must be improved. The text is understandable but there are many minor errors (too many to list).
· There are major omissions in discussion and citation of prior literature directly related to this study. Science builds upon prior science. Thus, it is not appropriate to ignore relevant prior literature. The reader should be led to understand how the current research builds upon the expanse of prior work.
The work by Stock’s group on EPN bacteria influence on ascaroside production was not discussed or cited (Roder et al. 2019). Also, given the theme of biological control using EPNs and ascarosides is prominent in this paper, it was very surprising that the impact of ascarosides on EPN dispersal and infectivity, and usage to directly enhance biocontrol efficacy was not mentioned; see wok from Shapiro-Ilan’s group (e.g., Perier et al. 2024; Oliveira-Hofman et al. 2019; Shapiro-Ilan et al. 2019), etc.
· Additionally, citations need to reflect the proper information for which they are cited. For example, citation number 11 (Bhat et al) is not appropriate for supporting the statement on lines 59-60. The reference only deals with distribution of EPNs (not nematodes as a whole).
· Line 19 and onward. It is not appropriate to refer to an organism by its genus name alone unless the genus itself is being discussed in context. Thus, referring to Galleria mellonella as just “Galleria” is not appropriate. After the species is fully spelled out the first time in abstract and main text then it should be referred to as G. mellonella. This is a mistake that needs correction in multiple places.
· Lines 64-64: EPNs also can enter through the insect cuticle (not only the natural openings), this should be corrected.
· Importantly, Some justification of the scope in the approach is warranted. First, why were only ascr#9 and ascr#12 chosen for the study as there are other ascarosides that are known to be associated and active for EPNs (only because they are the most abundant reported, but others may be very important too)? Secondly, why use only G. mellonella as a model. It is a very susceptible host and thus a host that is more resistant to EPN infection (and thus perhaps stronger immune system) may have been included for comparison. Third, although the assessment of gene expression in AMPs, ROS/RNS and LPS is very intriguing, why not also assess the level of encapsulation by hemocytes in infective juvenile nematodes (IJs) as a measure of immune response (in other words look at cellular immune response).
· The term melanization is not used correctly or at least departs substantially from its normal usage. Melanization is a series of chemical reactions that results in the production of darkened pigment around a wound or pathogen. The G. mellonella larvae are neither a wound or a pathogen, they are the host. Thus, melanization is often used in reference to entomopathogenic nematodes when indicating the response to IJ infection (immune response in terms of melanization of the IJs themselves). The term is not used to describe the color change in the host. The term should be changed to host color change or something similar instead of melanization.
In the first line of the discussion the authors equate the “melanization” color change in the host larvae to a measure of cellular immune response. A cellular immune response in insects should be measured internally, e.g., by hemocyte response, see above.
END
Comments on the Quality of English LanguageShould be improved. The text is understandable but there are many English grammar errors.
Author Response
For research article
|
Response to Reviewer X Comments
|
||
|
1. Summary |
|
|
|
Thank you very much for taking the time to review this manuscript. Please find the detailed responses below and the corresponding revisions/corrections highlighted changes in the re-submitted files.
|
||
|
2. Questions for General Evaluation |
Reviewer’s Evaluation |
Response and Revisions |
|
Does the introduction provide sufficient background and include all relevant references? |
Must be improved |
The introduction has been added more detail. |
|
Is the research design appropriate? |
Can be improved |
|
|
Are the methods adequately described? |
Yes |
|
|
Are the results clearly presented? |
Yes |
|
|
Are the conclusions supported by the results? |
Yes |
|
|
3. Point-by-point response to Comments and Suggestions for Authors |
||
|
Comments: English language writing must be improved. The text is understandable but there are many minor errors (too many to list). |
||
|
Response: We understand and respect the reviewer’s position regarding the assessment of the English quality in our manuscript. To address this concern, we’ve engaged a professional language editing service to improve the manuscript’s English quality.
|
||
|
Comments: There are major omissions in discussion and citation of prior literature directly related to this study. Science builds upon prior science. Thus, it is not appropriate to ignore relevant prior literature. The reader should be led to understand how the current research builds upon the expanse of prior work. Response: Thank you. We have added more detail and revised it to the discussion section.
|
||
|
Comments: The work by Stock’s group on EPN bacteria influence on ascaroside production was not discussed or cited (Roder et al. 2019). Also, given the theme of biological control using EPNs and ascarosides is prominent in this paper, it was very surprising that the impact of ascarosides on EPN dispersal and infectivity, and usage to directly enhance biocontrol efficacy was not mentioned; see wok from Shapiro-Ilan’s group (e.g., Perier et al. 2024; Oliveira-Hofman et al. 2019; Shapiro-Ilan et al. 2019), etc. Response : Thank you for the insightful feedback. I appreciate your attention to the relevant literature. In response, I will revise the manuscript to include a discussion of Stock’s group’s work on EPN bacteria and ascaroside production. Additionally, I’ll address the impact of ascarosides on EPN dispersal, infectivity, and biocontrol efficacy, drawing from studies by Shapiro-Ilan’s group in the discussion section.
Comments: Additionally, citations need to reflect the proper information for which they are cited. For example, citation number 11 (Bhat et al) is not appropriate for supporting the statement on lines 59-60. The reference only deals with distribution of EPNs (not nematodes as a whole). Response: Thank you so much. We have deleted it.
Comments: Line 19 and onward. It is not appropriate to refer to an organism by its genus name alone unless the genus itself is being discussed in context. Thus, referring to Galleria mellonella as just “Galleria” is not appropriate. After the species is fully spelled out the first time in abstract and main text then it should be referred to as G. mellonella. This is a mistake that needs correction in multiple places. Response: Thank you for pointing this out. The Galleria has changed to G. mellonella.
Comments: Lines 64-64: EPNs also can enter through the insect cuticle (not only the natural openings), this should be corrected. Response: Thank you. Done (can be found in Line 66-68)
Comments: Importantly, Some justification of the scope in the approach is warranted. First, why were only ascr#9 and ascr#12 chosen for the study as there are other ascarosides that are known to be associated and active for EPNs (only because they are the most abundant reported, but others may be very important too)? Secondly, why use only G. mellonella as a model. It is a very susceptible host and thus a host that is more resistant to EPN infection (and thus perhaps stronger immune system) may have been included for comparison. Third, although the assessment of gene expression in AMPs, ROS/RNS and LPS is very intriguing, why not also assess the level of encapsulation by hemocytes in infective juvenile nematodes (IJs) as a measure of immune response (in other words look at cellular immune response). Response: Thank you for pointing this out. First, The selection of ascr#9 and ascr#12 for our study was based on their abundant presence in EPNs, as reported in previous literature. While other ascarosides may indeed play important roles, our focus was on these two well-documented molecules. However, we appreciate the reviewer’s point, and future investigations could explore additional ascarosides to deepen our understanding. Secondly, The use of Galleria mellonella larvae as an infection model offers several advantages. These include cost-effectiveness, simplicity, temperature resemblance to the human body, and ethical considerations. While G. mellonella is susceptible, its practical benefits make it a valuable model. G. mellonella exhibits an innate immune system, enabling targeted research without the influence of adaptive immunity. However, including a more resistant host for comparison could enhance our understanding further.” Third, Thank you for the thoughtful question! Assessing the level of encapsulation by hemocytes in infective juvenile nematodes (IJs) is indeed a valuable approach to understanding the cellular immune response. While our study focused on gene expression related to AMPs, ROS/RNS, and LPS, incorporating hemocyte encapsulation as an additional measure could enhance the comprehensiveness of our investigation. Future research could explore this avenue to provide a more holistic view of the nematode immune response.
Comments: The term melanization is not used correctly or at least departs substantially from its normal usage. Melanization is a series of chemical reactions that results in the production of darkened pigment around a wound or pathogen. The G. mellonella larvae are neither a wound or a pathogen, they are the host. Thus, melanization is often used in reference to entomopathogenic nematodes when indicating the response to IJ infection (immune response in terms of melanization of the IJs themselves). The term is not used to describe the color change in the host. The term should be changed to host color change or something similar instead of melanization. Response: Thank you for sharing the reviewer comments. We have changed the word from “melanization” to “color change”.
Comments: In the first line of the discussion the authors equate the “melanization” color change in the host larvae to a measure of cellular immune response. A cellular immune response in insects should be measured internally, e.g., by hemocyte response, see above. Response: We have changed the word from “melanization” to “color change”.
4. Response to Comments on the Quality of English Language |
||
|
Point 1: Should be improved. The text is understandable but there are many English grammar errors. |
||
|
Response 1: We understand and respect the reviewer’s position regarding the assessment of the English quality in our manuscript. To address this concern, we have sought the assistance of a professional language editing service to ensure that the manuscript meets the required linguistic standards. We believe this will enhance the clarity and readability of our research for all readers
|
||
|
5. Additional clarifications |
||
|
- |
||

Reviewer 2 Report
Comments and Suggestions for Authors
Ascaroside pheromones are important molecules that play a key role in EPN's chemical signaling. Earlier studies indicated that ascaroside pheromones also increase EPN infectivity (invasion rate). However, here, The authors report that ascarosides enhance the G. mellonella immunity response and delay larval mortality. The results are interesting and can contribute to the field. However, the results should be discussed in detail with those studies on how the ascaroside accelerates mortality in one insect species while delaying it in other species. Although the authors have already touched on this topic briefly, this section needs to be expanded.
I have provided detailed comments on the manuscript.
There are also minor grammar and typing errors, please check carefully.

There are also minor grammar and typing errors. The manuscript needs to be checked.
Author Response
For research article
|
Response to Reviewer X Comments
|
||
|
1. Summary |
|
|
|
Thank you very much for taking the time to review this manuscript. Please find the detailed responses below and the corresponding revisions/corrections, which are highlighted in changes of the re-submitted files.
|
||
|
2. Questions for General Evaluation |
Reviewer’s Evaluation |
Response and Revisions |
|
Does the introduction provide sufficient background and include all relevant references? |
Yes |
|
|
Is the research design appropriate? |
Yes |
|
|
Are the methods adequately described? |
Yes |
|
|
Are the results clearly presented? |
Yes |
|
|
Are the conclusions supported by the results? |
Yes |
|
|
|
|
|
|
3. Point-by-point response to Comments and Suggestions for Authors |
||
|
Comments 1: However, the results should be discussed in detail with those studies on how the ascaroside accelerates mortality in one insect species while delaying it in other species. Although the authors have already touched on this topic briefly, this section needs to be expanded. |
||
|
Response 1: Thank you for pointing this out. We have discussed the studies on the effect of ascaroside on mortality, noting that “The beetles fed with ascaroside (asc-C9) have shown a significantly increased cold tolerance, as evidenced by a lower lethal temperature (LT50) compared to the control group. This suggests that asc-C9 contributes to enhancing the beetles’ cold tolerance and potentially delays mortality under colder environmental conditions (Zhang et al., 2020)” (Line 423-426).
Comments 1: I have provided detailed comments on the manuscript. Response 1: we have implemented the reviewer’s suggestion, which is highlighted in red in the revised manuscript.
|
||
|
4. Response to Comments on the Quality of English Language |
||
|
Point 1: There are also minor grammar and typing errors. The manuscript needs to be checked. |
||
|
Response 1: We understand and respect the reviewer’s position regarding the assessment of the English quality in our manuscript. To address this concern, we have sought the assistance of a professional language editing service to ensure that the manuscript meets the required linguistic standards. We believe this will enhance the clarity and readability of our research for all readers.
|
||
|
5. Additional clarifications |
||
|
- |
||

Reviewer 3 Report
Comments and Suggestions for Authors
I think the work is rather straightforward, and thus, only some editorial corrections would be necessary before the publication.
- L52: Reference should be indicated as reference number.
- L60: There should be more suitable reference to mention the abundance of nematodes.
- L117 (kayaii), 126 (Hererorhabditis), 131 (Galleria), 259 (Photorhabdus), 261 (G. mellonella), and 416 (G. mellonella) should be in italic.
- Figs 1-6: Resolution for the figures are not sufficient. For example, I could not read small characters in Fig. 5.
In addition to above editorial corrections, some researchers injected C. elegans and/or Oscheius spp. to G. mellonella (e.g., Ono et al., 2020: Parasitology 147, 279-286. doi: 10.1017/S0031182019001550, and many other works).
These works would be reviewed and mentioned in Discussion section.
Author Response
For research article
|
Response to Reviewer X Comments
|
||
|
1. Summary |
|
|
|
Thank you very much for taking the time to review this manuscript. Please find the detailed responses below and the corresponding revisions/corrections highlighted/in track changes in the re-submitted files.
|
||
|
2. Questions for General Evaluation |
Reviewer’s Evaluation |
Response and Revisions |
|
Does the introduction provide sufficient background and include all relevant references? |
Yes |
|
|
Is the research design appropriate? |
Yes |
|
|
Are the methods adequately described? |
Yes |
|
|
Are the results clearly presented? |
Yes |
|
|
Are the conclusions supported by the results? |
Yes |
|
|
3. Point-by-point response to Comments and Suggestions for Authors |
||
|
Comments: L52: Reference should be indicated as reference number. Response: Thank you so much. Done.
Comments: L60: There should be more suitable reference to mention the abundance of nematodes. Response: The sentence has been deleted.
Comments: L117 (kayaii), 126 (Hererorhabditis), 131 (Galleria), 259 (Photorhabdus), 261 (G. mellonella), and 416 (G. mellonella) should be in italic. Response: Thank you. Done.
Comments: Figs 1-6: Resolution for the figures are not sufficient. For example, I could not read small characters in Fig. 5. Response: We thank the reviewer for bringing this to your attention. Due to the issue with figure size, we have taken steps to ensure that all figures are now at the required resolution.
Comments: In addition to above editorial corrections, some researchers injected C. elegans and/or Oscheius spp. to G. mellonella (e.g., Ono et al., 2020: Parasitology 147, 279-286. doi: 10.1017/S0031182019001550, and many other works). These works would be reviewed and mentioned in Discussion section. Response: We appreciate the reviewer’s suggestion to include additional research works in our discussion. We have reviewed the mentioned studies, including Ono et al., 2020, and have incorporated these works into the discussion section to provide a more comprehensive analysis of the topic, noting that “The ingestion of haemocytes by C. elegans within the haemocoel of G. mellonella represents a significant finding, revealing a novel mechanism by which bacterial-feeding nematodes may evade host immune responses (Ono et al., 2020). (Line 337-339).
4. Response to Comments on the Quality of English Language |
||
|
Point 1: I am not qualified to assess the quality of English in this paper. |
||
|
Response 1: We understand and respect the reviewer’s position regarding the assessment of the English quality in our manuscript. To address this concern, we have sought the assistance of a professional language editing service to ensure that the manuscript meets the required linguistic standards. We believe this will enhance the clarity and readability of our research for all readers.
|
||
|
5. Additional clarifications |
||
|
- |
||

Reviewer 4 Report
Comments and Suggestions for Authors
The manucript “ Ascarosides and symbiotic bacteria of entomopathogenic nematodes regulate host immune response in Galleria mellonella larvae” is a well written manucript. The authors have used ascarosides and the bacterial symbiont to test the larval mortality of G. mellonella. Authors have carefullt tested the expression of of immune related genes and the melanization. I find the manuscript is ready for publication without any major changes. I have a few minor comments that authors can address before publications.
1. All the figures are blurr, not readable, they are not ready for publication or for anybody’s understnading.
2. Simple summary: In third line , change it into “In this study we provide…”
3. Change “…..and increasing related immune” to “…..and increasing immune- related”
4. Injection of Galleria Larvae: Correct the spelling of ascarosides.
5. Larval Mortality and Melanization: “After twelve hours of treatment, the dead larva body was examined visually to monitor the formation of melanin intensity, according to the degree of melanin synthesis; None, Light, Moderate and High. None=no visible signs of a melanin,…..” Break this sentence into two sentences”.
Author Response
For research article
|
Response to Reviewer X Comments
|
||
|
1. Summary |
|
|
|
Thank you very much for taking the time to review this manuscript. Please find the detailed responses below and the corresponding revisions/corrections highlighted/in track changes in the re-submitted files.
|
||
|
2. Questions for General Evaluation |
Reviewer’s Evaluation |
Response and Revisions |
|
Does the introduction provide sufficient background and include all relevant references? |
Yes |
|
|
Is the research design appropriate? |
Yes |
|
|
Are the methods adequately described? |
Yes |
|
|
Are the results clearly presented? |
Yes |
|
|
Are the conclusions supported by the results? |
Yes |
|
|
3. Point-by-point response to Comments and Suggestions for Authors |
||
|
Comments: All the figures are blur, not readable, they are not ready for publication or for anybody’s understnading. Response: We thank the reviewer for bringing this to your attention. Due to the issue with figure size, we have taken steps to ensure that all figures are now at the required resolution.
Comments: Simple summary: In third line, change it into “In this study we provide…” Response: Thank you so much. Done. (Line 330).
Comments: Change “…..and increasing related immune” to “…..and increasing immune- related” Response: Thank you so much. Done.
Comments: Injection of Galleria Larvae: Correct the spelling of ascarosides. Response: We thank the reviewer. Done.
Comments: Larval Mortality and Melanization: “After twelve hours of treatment, the dead larva body was examined visually to monitor the formation of melanin intensity, according to the degree of melanin synthesis; None, Light, Moderate and High. None=no visible signs of a melanin,…..” Break this sentence into two sentences”. Response: Done. The sentences have broken into “After 12 hours of treatment, the dead larva body was examined visually to monitor the formation of melanin intensity. The degree of melanin synthesis was classified according to [38]: None (no visible signs of melanin), Light (visible translucent melanin), Moderate (melanin with pigment semi-opaque to fully opaque), and High (melanin with strong pigmentation).” (Line 144-148)
4. Response to Comments on the Quality of English Language |
||
|
Point 1: English language fine. No issues detected. |
||
|
Response 1: We thank the reviewer.
|
||
|
5. Additional clarifications |
||
|
- |
||

Round 2
Reviewer 1 Report
Comments and Suggestions for Authors
My comments were addressed in a diligent manner.